# CEIR: Concept-based Explainable Image Representation Learning

## Abstract

In modern machine learning, the trend of harnessing self-supervised learning to derive high-quality representations without label dependency has garnered significant attention. However, the absence of label information, coupled with the inherently high-dimensional nature improves the difficulty for the interpretation of learned representations. Consequently, indirect evaluations become the popular metric for evaluating the quality of these features, leading to a biased validation of the learned representation's rationale. To address these challenges, we introduce a novel approach termed ***Concept-based Explainable Image Representation (CEIR)***. Initially, using the Concept-based Model (CBM) incorporated with pretrained CLIP and concepts generated by GPT-4, we project input images into a concept vector space. Subsequently, a Variational Autoencoder (VAE) learns the latent representation from these projected concepts, which serves as the final image representation. Due to the representation's capability to encapsulate high-level, semantically relevant concepts, the model allows for attributions to a human-comprehensible concept space. This not only enhances interpretability but also preserves the robustness essential for downstream tasks. For instance, our method exhibits state-of-the-art unsupervised clustering performance on benchmarks such as CIFAR10, CIFAR100, and STL10. Furthermore, capitalizing on the universality of human conceptual understanding, CEIR can seamlessly extract the related concept from open-world images without fine-tuning. This offers a fresh approach to automatic label generation and label manipulation.

## 1 Introduction

As self-supervised and unsupervised representation learning becomes more and more popular in real-world applications, the necessity for interpretable representations has attracted the attention of the research community recently Yang et al. (2022)Crabbé & van der Schaar (2022). Firstly, such interpretability aids users in understanding and navigating the feature space. Additionally, it enables the assessment of the quality and rationale behind the learned representations from a semantic standpoint. Secondly, this introduces alternative metrics for evaluating representation quality, deviating from the traditionally favored linear-probing evaluation protocol. The conventional linear probing method relies on labels to assign semantic information and evaluates the quality based on downstream task accuracy. Such methods, however, fall short of assuring the reliability and comprehensibility of the learned features.

In our pursuit of user-centric assessments of representation quality and reliability, we focus on discussing the post-hoc interpretation of representations—specifically, elucidating the contribution of defined inputs to representations after model training, commonly referred to as attribution. A significant proportion of deep learning's post-hoc interpretation techniques are designed for supervised contexts, allocating importance within the feature space according to predictions. Applying such methods directly onto self-supervised representations will face the challenge of the intricate high-dimensional spaces without ground-truth labels. Crabbé & van der Schaar (2022) employed surrogate functions to transfer attributions into a supervised context. On the other hand, investigations conducted by Kindermans et al. (2019) and Ghorbani et al. (2019a) underscored the instability and inconsistency of such feature-based methods. Moreover, they predominantly illuminate the significance within the raw input feature space, offering only limited insights when navigating complex interpretations. Meanwhile, example-based strategies Yang et al. (2022) presented contrastive ex-

amples with hinge upon the reference corpus, failing to deliver insights with finer granularity. Therefore, to ensure the validity of the attribution outcome, human concept-based interpretation is more appropriate since it can directly provide high-level semantic concepts with robustness.

A growing area of research aims to provide explanations based on human concepts. Koh et al. (2020)Kim et al. (2018)Oikarinen et al. (2023)Ghorbani et al. (2019b). Compared with feature-based approaches, concept-based models try to unveil concepts inherently tied to model outputs. When fed with an input instance paired with corresponding model output, these methods provide elucidate contributions from either fine-grained or coarse-grained concepts. Kim et al. (2018) employs user-defined concepts to bridge the gap between learned features and the conceptual space, thus necessitating concept-related supervisory signals and requiring the creation of concept-based datasets and subsequent pretraining to obtain vectors pertinent to user-defined explanations. An alternate trajectory, rooted in the Concept Bottleneck Model (CBM) Koh et al. (2020)Oikarinen et al. (2023), primarily emphasizes projecting hidden, learned feature vectors onto a blend of concept vectors using a concept bottleneck layer. Yet, its adaptability remains confined to supervised learning contexts. Apart from the above methods, the automated explanation technique Ghorbani et al. (2019b) provides concept-centric explanations under unsupervised regimes, devoid of explicit concept-label dependencies. These explanations, however, emerge from the input feature space itself, failing to ascend to text-described abstract concepts. Despite the impressive strides in concept-based interpretative techniques, crafting interpretations of latent representations using human-interpretable concepts, especially in the absence of annotated labels, remains a challenging problem.

To meet the gap, we propose ***"CEIR: Concept-based Explainable Image Representation Learning"***—a novel representation learning method that introduces the concept bottleneck model into the domain of representation learning. Our extensive empirical experiments demonstrate that CEIR adeptly captures semantically enriched representations. Concurrently, these representations can be attributed to the concept space, providing intuitive and human-centric explanations of the learned representations. An auxiliary advantage of CEIR, rooted in harnessing the ubiquity of human conceptual understanding, lies in its prowess to derive concepts from open-world images without fine-tuning. This innovation paves the way for automated label generation and label manipulation.

- We introduce CEIR, a novel image representation learning method, adept at harnessing human concepts to bolster the semantic richness of image representations.

- Demonstrating its efficacy, CEIR achieves state-of-the-art results in unsupervised clustering task on benchmarks including CIFAR10, CIFAR100, and STL10. This underscores its capability to encapsulate the semantic property of input features intertwined with diverse concepts.

- CEIR allows interpretation incorporated with label-free attribution methods Crabbé & van der Schaar (2022), providing users with a coherent and valid concept-driven interpretation, facilitating the assessment of the learned representation's quality and reliability.

## 2 RELATED WORK

Most existing interpretation methods rely on ground-truth labels to clarify their prediction. However, recent discoveries present label-free interpretation techniques using both supervised and unsupervised learning, providing a promising concept-based approach to enhance model comprehensibility for humans.

### 2.1 CONCEPT BASED EXPLANATION

Various research aiming at testing with Concept Activation Vectors (T-CAV) has enabled the extraction of high-level and interpretable concepts. Kim et al. (2018) leverages the manual-crafted concepts for categories and applies quantitative methods to reveal explainable concepts. On top of that, the Automatic Concept-Based Explanation Ghorbani et al. (2019b) takes the image segments under different resolutions as input, using the super-pixel segmentation and clustering approaches to extract the concepts. It returns the importance scores through the similar computation process of T-CAV. In parallel, some other works try to disentangle the concept into another feature space whose dimensions are unrelated Chen et al. (2020c), like beta-TC-VAE Chen et al. (2018). This technique

can also be applied to the semantic segmentation tasks to increase the model's generalization ability Choi et al. (2021).

## 2.2 EXPLAINABILITY FOR UNSUPERVISED REPRESENTATION

In the field of label-free settings, the latent space features are interpreted using feature attribution methods. The label-free Feature Importance Crabbé & van der Schaar (2022) evaluates the contribution of features in latent space concerning the representation vectors. This method first takes the dot product of the representation and then employs the previous supervised attribution methods to attribute this product to the input feature. In addition, Yang et al. (2022) proposed to explain representations learned by contrastive learning with class-preserving features in image augmentations. The method uses contrastive similarity between visual features and corpus to construct a more meaningful support set Lin et al. (2022); meanwhile, it enforces the visual features to be dissimilar from the foil sample.

## 2.3 CONCEPT BOTTLENECK MODEL

Under the umbrella of the Concept Based Explanation, the Concept Bottleneck Model (CBM) Koh et al. (2020) aims to utilize the bottleneck model to project the input features into concept space. More specifically, CBM uses supervised training to learn the mapping from raw inputs to high-level explainable concepts, requiring supervised labels for training and concept annotation for establishing concept space. Recently, with the evolution of large-scale pre-trained models such as CLIP Radford et al. (2021), Oikarinen & Weng (2022)Yang et al. (2023) utilize share text-image embedding space to align representation between text concept and input image. The Large Language Model (LLM) is introduced for automatic concept generation to eliminate the requirement of concept annotation. For example, Oikarinen et al. (2023) proposes Label-free CBM (LF-CBM), which leverages GPT-3 Brown et al. (2020) for concept generation and trains a concept-aware classifier on concept vector supervisely. However, the aforementioned methods either need supervised labels or face restrictions on a pretraining dataset that cannot be scaled to arbitrary open-world images.

## 3 METHOD

In the concept-based explanation model, we need to define the reference concept set for the model to inquire and get the related concept outputs. Therefore, the first part of our method is constructing a meaningful and diverse concept set. After obtaining the concept set, we utilize the same training method for the concept bottleneck layer in the LF-CBM to construct the corresponding concept vector for each image. Then, we can get a representation feature vector for the input image. Furthermore, we utilize the VAE (Variational AutoEncoder) Kingma & Welling (2013) to conduct the reconstruction task and utilize the latent vector $h$ in VAE as our final image representation.

### 3.1 PROMPT-BASED CONCEPT POOL GENERATION

Large language models have enabled the capability of exploring the potential concepts associated with visual objects. To obtain user-interpretable concepts for various domains of images, we utilize the GPT model for querying several levels of vision concepts for the given image benchmarks. Similar to LF-CBM, the concept pool generation consists two stages: (1) Concept querying (2) Concept filtering. Unlike the previous work, we instead utilize GPT-4 with role-based prompts White et al. (2023) to drive the concept querying. Moreover, due to the nature of the unsupervised task, the ground-true label is inaccessible; we further modify the filtering stage by adding class-related concepts on purpose to facilitate the emergence of optimal concept vectors for the input images. The ablation in Appendix A.2.3 shows the superiority of preserving class-related concepts, whereas a slight decay of performance by removing such concepts doesn't hurt the overall good performance, suppressing the concerns of label leakage. The details of the prompt and examples are provided in Appendix A.1 as well.

Figure 1: The schematic representation of the CEIR pipeline. Phases 1 and 2 of the training steps are shaded in different colors.

## 3.2 CONSTRUCTING CONCEPT VECTORS

Upon procuring the reference concept set, our subsequent task is to derive the linear project layer $W_c \in \mathbf{R}^{M \times d_0}$ where $d_0$ is the latent dimensionality of the backbone and $M$ is the number of the reference concept, transforming the image backbone output into the concept vector $q_i \in \mathbf{R}^M$ where $i \in [1, ..., N]$ and N denotes the input instance number. Here, every dimension of $q_i$ uniquely denotes a concept from the reference set, with the magnitude indicating the correlation between the concept and its corresponding instance. A pivotal aspect of this approach is leveraging the multi-modal prowess of CLIP, which adeptly bridges natural language concepts and images. This follows the method in the LF-CBM, leveraging the CLIP-Dissect, given our two mapping domains—input data $X = \{x_1, ..., x_N\}$ and the reference data $C = \{t_1, ...t_M\}$. We harness the pretrained CLIP text encoder $E_T$ and image encoder $E_I$ to generate the similarity matrix $P_{i,j} = E_I(x_i) \cdot E_T(t_j)$. With CLIP's capability to indicate semantic similarities across images and text, our central aim is to employ the similarity information present in CLIP's embedding space as weak supervision, guiding the concept bottleneck layer to mirror this understanding. Equipped with the concept bottleneck layer $(W_c)$, we aspire for the projection $q_i$ to precisely encapsulate the interrelation with concepts. Thus, for each concept(dimension) $k$ where $k \in [1, ..., M]$ we utilize the $l_k = [q_{1,k}, q_{2,k}, ..., q_{N,k}]^T$ to represent the activation of the concept k in all input instances and align $l_k$ with the similarity score in the same concept in similarity matrix $P_{:,k}$. This rationale leads us to utilize this loss function tailored to synchronize the conceptual vectors with the similarity matrix:

$$L(W_c) = \sum_{k=1}^{M} -\text{sim}(t_k, l_k) := \sum_{k=1}^{M} -\frac{\bar{l}_k^3 \cdot P_{:,k}^3}{\left\| \bar{l}_k^3 \right\|_2 \left\| P_{:,k}^3 \right\|_2}. \tag{1}$$

where $\overline{l_k}$ denotes zero-mean and one-standard deviation normalized $l_k$.

## 3.3 LEARNING THE LATENT REPRESENTATION OF CONCEPT VECTOR

Having established a projection from the input image space to the concept vector space—the output of the concept bottleneck layer $q_i$—we've essentially mapped images to an embedding space. In

this space, each dimension corresponds to a distinct human-understandable concept. However, due to their high dimensionality and trivial value variations, these concept vectors don't readily serve as direct representations. To address this, we employ a VAE to distill the high-dimensional concept vector $q_i$ into a more manageable low-dimensional representation $h_i \in \mathbf{R}^K$ where K denotes the latent dimensionality of the VAE with encoder $f$ and decoder $g$, we adopt the vanilla VAE model, focusing on reconstruction as the primary objective.

$$L_{\text{VAE}}(f, g) = \sum_{i=1}^{N}(||q_i - g(f(q_i))||^2 + \text{KL}(f(q_i)))$$ (2)

where KL denotes the KL divergence between the latent space and the standard normal distribution.

### 3.4 LABEL-FREE ATTRIBUTION FOR CONCEPT VECTOR

Upon obtaining the latent representation $h_i$ for each image $x_i$, we need to attribute the model's output back to the input concept vector space, ensuring a concept-driven post-hoc interpretation. However, our approach has a notable difference from previous works. Instead of focusing on predictions, our model's output is a learned representation. This shift necessitates a change in our interpretive process. Instead of elucidating the relationship between the label and the instance, we aim to illuminate the composition of the instance itself. Fortunately, the label-free explanation method Crabbé & van der Schaar (2022) offers a roadmap. By leveraging this method, we can attribute the high-dimensional features extracted from unsupervised learning approaches. Some adjustments are required, particularly when integrating with supervised attribution methods. This process begins with the VAE's encoder. $f : \mathcal{Q} \rightarrow \mathcal{H}$ maps the concept space into the latent space. We can attribute each input concept vector dimension $q_i$ to get the importance score $b_{i,j}$ where $j \in [1, ..., K]$. This score reflects the importance of each concept in $q_i$ in assigning the representation $h_i = f(q_i)$. We utilize the Integrated Gradient Qi et al. (2019): $IG_j(\cdot, \cdot) : \mathcal{A}(\mathbb{R}^{\mathcal{Q}}) \times \mathcal{Q} \rightarrow \mathbb{R}$ with the full-zero input concept vector baseline to construct the attribution function, where $\mathcal{A}(\mathbb{R}^{\mathcal{Q}})$ denote hypothesis set of the function maps the input space $\mathbf{Q}$ to the scalar output $\mathcal{R}$.

$$b_{i,j} \equiv IG_j(g_{\boldsymbol{q_i}}, \boldsymbol{q_i})$$ (3)

$$\forall \tilde{\boldsymbol{q}} \in \mathcal{Q} : g_{\boldsymbol{q}}(\tilde{\boldsymbol{q}}) = \langle \boldsymbol{f}(\boldsymbol{q}), \boldsymbol{f}(\tilde{\boldsymbol{q}}) \rangle_{\mathcal{H}}.$$ (4)

where $\langle \cdot, \cdot \rangle_{\mathcal{H}}$ denotes an inner product for the space $\mathcal{H}$. Then, we can get an importance score for each concept based on the attrition results, and we can get the final weighted concept vector $q_i^* = [b_{i,1}q_{i,1}, ..., b_{i,K}q_{i,K}]^T$. Compared with the original concept vectors, these importance-weighted vectors exhibit more discernible significance, thereby making the differences between concepts more pronounced. The reason is that the representation $h$ can be regarded as a compress of the concept vector, it needs to capture the saliency of each concept for the input. This enforces the VAE to enhance the pertinent concepts while possibly diminishing others that are less crucial.

## 4 EXPERIMENT

CEIR offers two major strengths: (1) generating high-quality concept-based representation and (2) human-interpretable concepts from the underlying concept space, all achieved in an unsupervised manner. In this section, we first quantitatively evaluate the quality of the produced representation $h$ by performing image clustering and linear probing on CIFAR10, CIFAR100-20, CIFAR100 Krizhevsky et al. (2009), STL10 Coates et al. (2011), and ImageNet Deng et al. (2009). The key details are exhibited in Table 1, where the selected benchmarks contain a diversity of categories, resolutions, and dataset scales. We compare our approach against the state-of-the-art clustering and representation learning models, as well as against standard supervised learning baselines. Furthermore, we evaluate CEIR on both the training benchmark and unseen open-world images, demonstrating its adaptability in extracting explainable visual concepts in the format of plain text without annotation or prerequisite knowledge.

Table 1: Specifications and partitions of selected datasets. * denotes additional unlabeled samples.

| Dataset | Image size | # Training | # Testing | # Classes |
|---|---|---|---|---|
| CIFAR10 | $32 \times 32$ | 50,000 | 10,000 | 10 |
| CIFAR100-20 | $32 \times 32$ | 50,000 | 10,000 | 20 |
| CIFAR100 | $32 \times 32$ | 50,000 | 10,000 | 100 |
| STL10 | $96 \times 96$ | 500 + 100,000* | 800 | 10 |
| ImageNet | $224 \times 224$ | 1,281,167 | 50,000 | 1,000 |

## 4.1 UNSUPERVISED CLUSTERING

We evaluate the efficacy of the concept-based latent embedding within the domain of unsupervised clustering. Specifically, for datasets CIFAR10, CIFAR100-20, CIFAR100, and STL10, we employ three different pretrained backbone models: CLIP-RN50, CLIP-ViT-B/16, and CLIP-ViT-L/14 Radford et al. (2021). For the ImageNet, we only load the ResNet50 He et al. (2016). Our quantitative assessments are grounded in three primary criteria: (1) Normalized Mutual Information (NMI), (2) Clustering Accuracy (ACC), and (3) Adjusted Random Index (ARI). To optimize the training of the concept projection layer, we adopt the methodology proposed by Oikarinen et al. (2023), wherein the testing set is periodically assessed for early stopping. In our VAE model training, we merge training and testing sets, a decision to enhance latent representation learning. We extract the VAE's latent embedding $h$ and apply K-means on $h$ produced from the testing set for the clustering process. For comparison, we've chosen the baseline models from three distinct paradigms: clustering, contrastive representation learning, and supervised learning. Specifically for the representation learning models, we initialize them with pre-trained weights and subsequently employ K-means clustering on the generated latent embeddings from the testing set. This follows the same settings as ours for fair comparison. Table 2 presents a detailed breakdown of our clustering results. The settings of the experiments and ablation experiments are provided in Appendix A.2.

Table 2: Image clustering performance on selected benchmarks. The results of ImagNet were collected on the validation set. For backbone models, the TEMI model employs pre-trained ViT-B/16, SimCLR, and MoCoV2 integrated ResNet50 model. † represents the training procedure, including additional data (e.g., testing set). * represents applying K-means on the testing set. The best results are shown in boldface. N/A indicates the experiment was not conducted on the corresponding dataset.

| Datasets | CIFAR10 | | | CIFAR100-20 | | | CIFAR100 | | | STL10 | | | ImageNet | | |
|---|---|---|---|---|---|---|---|---|---|---|---|---|---|---|---|
| Methods (%) | NMI | ACC | ARI | NMI | ACC | ARI | NMI | ACC | ARI | NMI | ACC | ARI | NMI | ACC | ARI |
| *unsupervised clustering* | | | | | | | | | | | | | | | |
| K-Means† MacQueen et al. (1967) | 8.70 | 22.90 | 4.90 | 13.00 | 8.40 | 2.80 | N/A | N/A | N/A | 12.50 | 19.20 | 6.10 | N/A | N/A | N/A |
| VAE† Kingma & Welling (2013) | 24.50 | 29.10 | 16.70 | 10.80 | 15.20 | 4.00 | N/A | N/A | N/A | 20.00 | 28.20 | 14.60 | N/A | N/A | N/A |
| DCGAN† Radford et al. (2015) | 26.50 | 31.50 | 17.60 | 12.00 | 15.10 | 4.50 | N/A | N/A | N/A | 21.00 | 29.80 | 13.90 | N/A | N/A | N/A |
| DAC† Chang et al. (2017) | 39.59 | 52.18 | 30.59 | 18.52 | 23.75 | 8.76 | N/A | N/A | N/A | 36.56 | 46.99 | 25.65 | N/A | N/A | N/A |
| CC† Li et al. (2021) | 70.50 | 79.00 | 63.70 | 43.10 | 42.90 | 26.60 | N/A | N/A | N/A | 76.40 | 85.00 | 72.60 | N/A | N/A | N/A |
| SCAN Van Gansbeke et al. (2020) | 79.70 | 88.30 | 77.20 | 48.60 | 50.70 | 33.30 | N/A | N/A | N/A | 69.80 | 80.90 | 64.60 | 72.00 | 39.90 | 27.50 |
| SPICE† Niu et al. (2022) | 86.50 | 92.60 | 85.20 | 56.70 | 53.80 | 38.70 | N/A | N/A | N/A | 87.20 | 93.80 | 87.00 | N/A | N/A | N/A |
| ProPos† Huang et al. (2022) | 88.60 | 94.30 | 88.40 | 60.60 | 61.40 | 45.10 | N/A | N/A | N/A | 75.80 | 86.70 | 73.70 | N/A | N/A | N/A |
| TEMI Adaloglou et al. (2023) | 88.60 | 94.50 | 88.50 | 65.40 | **63.20** | **48.90** | 76.90 | **67.10** | 53.30 | 96.50 | 98.50 | 96.80 | **81.40** | **58.00** | **45.90** |
| *representation learning* | | | | | | | | | | | | | | | |
| SimCLR* Chen et al. (2020a) | 11.21 | 20.95 | 11.81 | 11.82 | 15.39 | 3.71 | 24.95 | 9.65 | 2.49 | 52.94 | 53.94 | 34.40 | 41.52 | 4.27 | 0.27 |
| MoCoV2* Chen et al. (2020b) | 26.46 | 34.18 | 14.26 | 22.90 | 23.74 | 7.96 | 33.42 | 15.71 | 5.54 | 59.52 | 58.68 | 34.99 | 63.50 | 24.74 | 11.33 |
| CLIP (ResNet50)* | 48.62 | 54.97 | 35.57 | 36.89 | 35.75 | 18.86 | 45.07 | 25.20 | 12.80 | 86.09 | 89.81 | 80.80 | 68.08 | 31.77 | 18.62 |
| CLIP (ViT-B/16)* | 75.78 | 78.25 | 68.94 | 50.84 | 48.90 | 31.72 | 61.64 | 45.27 | 30.12 | 93.54 | 94.94 | 90.52 | 75.49 | 44.63 | 32.39 |
| CLIP (ViT-L/14)* | 83.67 | 83.38 | 78.20 | 49.18 | 44.17 | 30.35 | 66.69 | 51.32 | 38.24 | 95.13 | 95.76 | 92.10 | 79.68 | 52.02 | 40.81 |
| ***ours*** | | | | | | | | | | | | | | | |
| CEIR (ResNet50)†* | 53.27 | 69.19 | 45.31 | 44.81 | 46.93 | 26.90 | 52.45 | 36.26 | 21.55 | 89.71 | 95.24 | 89.87 | 78.26 | 53.09 | 38.11 |
| CEIR (ViT-B/16)†* | 81.36 | 90.46 | 80.03 | 55.63 | 57.33 | 38.80 | 67.27 | 54.90 | 38.42 | 96.19 | 98.48 | 96.66 | N/A | N/A | N/A |
| CEIR (ViT-L/14)†* | **90.08** | **95.70** | **90.71** | **65.91** | 62.53 | 48.26 | **78.04** | 66.77 | **54.25** | **97.87** | **99.19** | **98.21** | N/A | N/A | N/A |
| *supervised learning* | | | | | | | | | | | | | | | |
| ResNet50 He et al. (2016) | | 94.78 | | | 86.21 | | | 77.39 | | | 89.13 | | | 75.30 | |

In a thorough comparison with clustering and representation learning models, our pipeline, utilizing the different backbone models, consistently demonstrates marked superiority across various evaluation metrics. On smaller datasets like CIFAR10 and STL10, CEIR not only surpasses the performance of established unsupervised clustering methods such as TEMI Adaloglou et al. (2023) and representation learning models like CLIP(ViT-L/14), but also sets new state-of-the-art benchmarks. For instance, on CIFAR10, CEIR(ViT-L/14) achieves an impressive NMI of 90.08%, ACC of 95.70%, and ARI of 90.71%, all of which are the highest reported scores for this dataset. For STL10, by incorporating an additional 100,000 unlabeled images during training, we achieved performance surpassing both prior methods and supervised learning baselines. The effect of adding unlabeled images to CEIR is available in Appendix A.3. Furthermore, while ImageNet poses challenges due to its large scale and diversity, CEIR still manages to secure tier 1 performance, challenging and in some cases, surpassing the results of renowned models in the domain. In Figure 2, we present the t-SNE visualization derived from the $h$ of CEIR on the CIFAR10, where all three underlying backbone models demonstrate well-dispersed clustering outcomes. Remarkably, these achievements are even more notable when we consider that CEIR only integrates one trainable projection layer (MLP) and a shallow VAE consisting of a two-layer MLP.

Figure 2: Visualization of t-SNE clustering of $h$ when utilizing different backbones on CIFAR10. Individual points correspond to samples within the embedding space, with their colors indicating the actual ground-truth labels. For clarity, a subset of 1000 samples is randomly selected for each label.

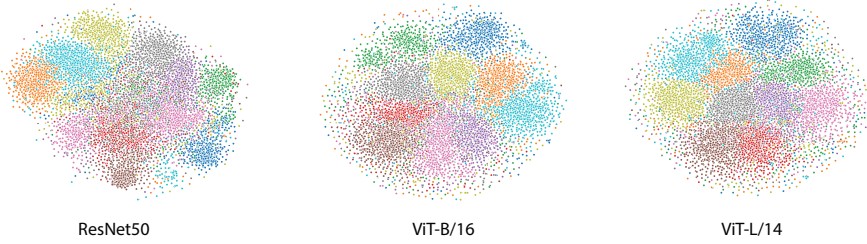

ResNet50         ViT-B/16         ViT-L/14

## 4.2 Linear Probing

Here, we conduct linear probing in a supervised manner where a linear layer is trained on top of the latent embedding to examine the discriminative ability and representational quality of the embedded features. The training is conducted on each benchmark's training set, and the evaluation metrics are evaluated on the testing set. For representation learning models, pre-trained weights are loaded. Table 3 offers the details of the results. Compared with the CLIP, our pipeline indicates a slight reduction in performance based on the provided evaluation metrics. Focusing on the specifics, there's a more noticeable decrease with the ResNet50 backbone, while the ViT backbone sees only a minor drop. This outcome is reasonable considering the primary intent of our pipeline, which is to transform the latent embeddings produced by the backbone model into a more human-interpretable concept space. Though this transformation enhances interpretability, it can reduce the representative capability of the original embeddings, potentially leading to some information loss. Yet, looking at the broader picture, our pipeline's results are still quite promising. Particularly when set against representation learning models like MoCo and SimCLR, our methodology consistently delivers superior results, highlighting its effectiveness.

## 4.3 Learn Concept based Explainable latent representation

What makes CEIR distinct in the landscape of unsupervised representation learning is its unique capability to produce human-interpretable concepts from the learned representations. Illustrating this, Figure 3 visualizes decoded concepts from two sets of images. The left panel displays pairs of similar categories "Golf Cart" vs. "Sports Car" and "Leopard" vs. "Lion" from the ImageNet training set on which the model is trained. By setting a threshold at 0.05, based on the product of the feature importance score and concept activation vector, we select the primary concepts for the target image. The term "Not" signifies a negative value in the concept activation vector, indicating dissimilarity in the concept space. In the depicted alluvial maps, the height of each concept reflects

Table 3: Linear probing. The result of DINO(ViT-B/16) refers to Adaloglou et al. (2023)

| Datasets | CIFAR10 | | | CIFAR100-20 | | | STL10 | | |
|---|---|---|---|---|---|---|---|---|---|
| Methods (%) | NMI | ACC | ARI | NMI | ACC | ARI | NMI | ACC | ARI |
| ***Ours*** | | | | | | | | | |
| CEIR (ResNet50) | 69.44 | 82.93 | 66.92 | 60.45 | 71.37 | 51.96 | 90.99 | 96.11 | 91.65 |
| CEIR (ViT-B/16) | 87.24 | 94.21 | 87.71 | 72.72 | 81.42 | 66.53 | 96.98 | 98.85 | 97.47 |
| CEIR (ViT-L/14) | 93.04 | 97.19 | 93.89 | 80.37 | 87.70 | 76.61 | 98.31 | 99.40 | 98.67 |
| *representation learning* | | | | | | | | | |
| CLIP (ResNet50) | 77.57 | 88.26 | 76.35 | 64.68 | 75.31 | 57.18 | 93.49 | 97.22 | 94.00 |
| CLIP (ViT-B/16) | 90.63 | 95.94 | 91.28 | 80.38 | 87.73 | 76.58 | 97.80 | 99.20 | 98.24 |
| CLIP (ViT-L/14) | 95.14 | 98.11 | 95.14 | 85.53 | 91.67 | 83.49 | 99.37 | 99.79 | 99.53 |
| DINO (ViT-B/16) | 92.50 | 96.80 | 93.10 | 82.40 | 89.50 | 79.50 | 97.80 | 99.20 | 98.20 |
| SimCLR (ResNet50) | 53.54 | 74.39 | 57.01 | 39.88 | 54.26 | 30.03 | 70.47 | 82.60 | 67.03 |
| MoCov2 (ResNet50) | 59.63 | 77.44 | 57.24 | 49.44 | 64.69 | 41.31 | 81.48 | 91.15 | 81.53 |

its importance. We observe that both image pairs share high-level concepts like "vehicle" and "big cat", whereas they also preserve unique semantic elements. For example, the image of the sports car is characterized by "rear hatchback" and "blue and white exterior", while the leopard has "orange and black patterns" and the lion is noted for its "lion-like mane". Additionally, background concepts like "fairway" and "African savannah" emerge, which are often ignored in typical image annotations. Furthermore, many secondary concepts such as "safari vehicle" and "cheetah" are raised, suggesting potential categories that are in close semantic similarity to the target class. All of this evidence indicates that CEIR can capture both high-level and fine-granted semantic concepts.

To explore the generalizability of CEIR, we adopt the model trained on ImageNet to arbitrary real-world images. As depicted in the right panel of Figure 3, CEIR proficiently identifies semantic concepts across varying scales, from specific elements like "basketball" and "umbrella" to broader themes such as "university" and "campsite". Remarkably, concepts such as the dressing of individuals, the hot spring in the background, and the Venice style of architecture also emerge, showcasing its ability to uncover contextual information from unseen open-world images.

Figure 3: Visual representations of concepts derived from images. The left panel presents images from two similar categories within ImageNet, while the right panel exhibits arbitrary real-world images alongside their revealed concepts. The model was trained on Imagent using the ResNet50 backbone.

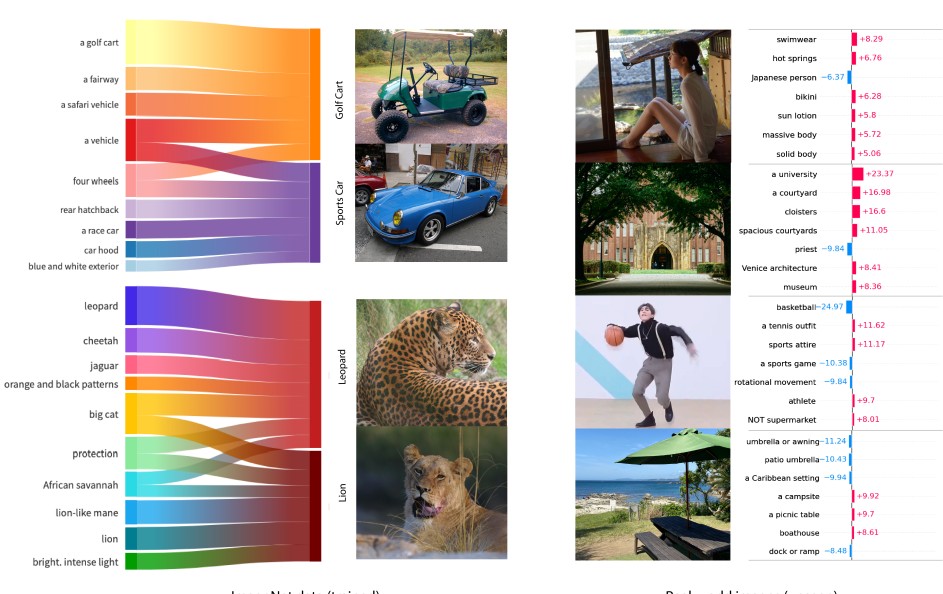

ImageNet data (trained)                    Real-world images (unseen)

### 4.4 OPEN-WORLD CONCEPT MINING AND LABEL GENERATION

The ability of CEIR to uncover human-interpretable concepts without annotations has made it a promising approach in various scenarios. For example, they can be integrated as auxiliary features, providing contextual information to enhance the latent embeddings in other models. In this demonstration, we apply the CEIR pipeline for open-world data mining to build a toy dataset from scratch. We gathered 24 images from the Internet, all related to the keyword "Kamakura", and processed them through our pipeline. Figure 4 displays some of the sourced images and the identified concepts. The terms in the word cloud give a broader overview of the main themes within our new dataset, allowing for an understanding of the primary semantic elements present. Moreover, the identified concepts for each image can serve as multi-class labels, preparing them for subsequent downstream tasks. This illustrates the capability of CEIR in facilitating automated multi-class label generation, as well as potential label manipulation tasks. An alignment score, drawn between manually crafted labels and those generated automatically, can act as a metric to identify potential false label assignments.

Figure 4: Examples of open-world concept mining. The set of images on the right-hand side were sourced from the Internet, capturing scenes from Kamakura, a town located south of Tokyo, Japan. A total of 24 images were collected and subsequently processed through the CEIR pipeline, which was trained on ImageNet. The word cloud depicted on the left offers a visualization of the uncovered concepts from this toy dataset. The front size of each term in the word cloud represents its frequency of occurrence. The threshold was set to 5 while filtering the activated concepts.

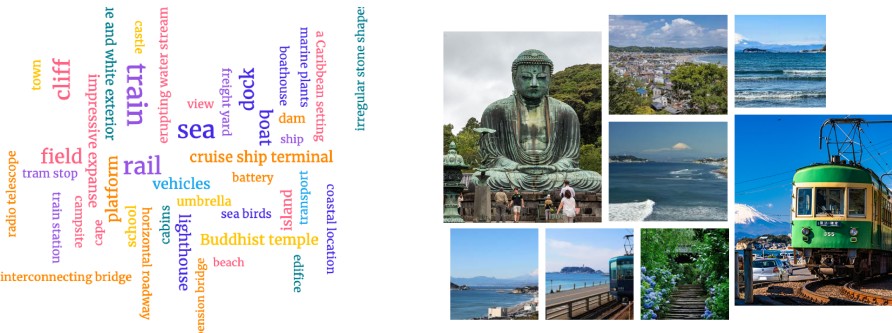

## 5 DISCUSSION

In our newly proposed CEIR approach, we leverage the VAE for a self-supervised task, aiming to learn low-dimensional representations of concept vectors. While our current framework is effective, there's room for incorporating more advanced self-supervised methods like contrastive learning to derive even more robust representations.

One limitation of our method hinges on the generated concept reference set. However, this also offers a significant degree of flexibility, allowing for user customization, especially when aided by a language model. This capability could be invaluable for generating finer-grained, multi-class labels for unlabeled datasets. Furthermore, it paves the way for researchers to delve deeper into features, potentially discerning causal from spurious concepts associated with images.

A noteworthy characteristic of these concepts is their resilience against domain shifts. Since such shifts seldom impact the primary, contributory concepts significantly, the representations derived through our approach might exhibit strong potential in domain generalization or adaptation tasks. These avenues offer promising directions for future research.

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

## A  APPENDIX

### A.1  CONCEPT GENERATION

The process of concept generation is twofold: initial concept set formation and subsequent refinement. We leverage GPT, available through OpenAI's API, given its established capability in extracting extensive domain knowledge. When tasked with a specific benchmark, GPT provides visual descriptions of categories, which then assist in generating class-specific attributes. These attributes consolidate into an initial set of concepts. To enhance the utility and specificity of this set, a filtering process is employed. This involves trimming extensive concepts, removing redundancy among them, validating their relevance against the training data, and verifying their representational capability during training. The outcome is a streamlined and interpretable set of concepts primed for subsequent analyses. Note that for the CIFAR100-20 benchmark, we leverage both the coarser and fine labels to generate a more representative concept set.

#### A.1.1  GPT-4 CONCEPT GENERATION

Unlike LF-CBM, which uses GPT-3, we prompt GPT-4 for more precise concept generation. As visualized in Figure 5, the queries consist of three chat completion tasks defined under a specific system role, which retrieves the visual descriptions from different aspects.

Figure 5: Prompts used to query GPT-4 for visual concept generation.

Role

> You are an assistant that provides visual descriptions of objects. Use only adjectives and nouns in your description. Ensure each description is unique, short, and direct. Do not use qualifiers like 'typically', 'generally', or similar words.

Chat completion

> List the most important features for recognizing something as a \"goldfish\":\n\n-bright orange color\n-a small, round body\n-a long, flowing tail\n-a small mouth\n-orange fins\n\nList the most important features for recognizing something as a \"beerglass\":\n\n-a tall, cylindrical shape\n-clear or translucent color\n-opening at the top\n-a sturdy base\n-a handle\n\nList the most important features for recognizing something as a \"{}\":

> Give superclasses for the word \"tench\":\n\n-fish\n-vertebrate\n-animal\n\nGive superclasses for the word \"beer glass\":\n\n-glass\n-container\n-object\n\nGive superclasses for the word \"{}\":

> List the things most commonly seen around a \"tench\":\n\n- a pond\n-fish\n-a net\n-a rod\n-a reel\n-a hook\n-bait\n\nList the things most commonly seen around a \"beer glass\":\n\n- beer\n-a bar\n-a coaster\n-a napkin\n-a straw\n-a lime\n-a person\n\nList the things most commonly seen around a \"{}\":

Answer

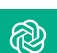

> -aerodynamic shape\n -jet engines \n - large wings \n - wheels \n - vehicle, ...

#### A.1.2  COMPARISON WITH GPT-3

Compared with GPT-3, the advantages of GPT-4 involve generating more accurate, natural, and diverse descriptions for a class to serve as concepts. This may include a better understanding of a class because GPT-4 has a broader knowledge base, increased model capacity, and more training data. Also, GPT-4 can generate a larger concept bank for a single class. For example, given CIFAR-10 datasets, GPT-3 generates 143 concepts while GPT-4 generates 182 concepts. Figure 6 displays the examples of concepts set produced by GPT-3 and GPT-4.

#### A.1.3  CONCEPT FILTERING

While the filtering procedure takes inspiration from LF-CBM, a crucial deviation is our approach toward class-related concepts. Rather than eliminating them, we intentionally retain and tag them

Figure 6: Concept sets produced for selected categories in CIFAR10.

with the class label. The comparative experiments can be found in Ablation A.2.3. Below are the steps for filtering:

I. Regarding concept length, we eliminate concepts exceeding 30 characters to maintain simplicity and avoid unnecessary complexities.

II. We are comfortable with our concept bottleneck layer, including the output classes themselves. We intentionally preserve class-related concepts and add class tags for a more representative concept vector generation.

III. To prevent the presence of redundancy or synonyms, we remove overly similar concepts. Using the same embedding space as previously mentioned, we exclude any concept that exhibits a cosine similarity of more than 0.9 with another concept already in the concept set.

IV. We erase concepts from the output of the concept bottleneck layer that is not explainable. This is done by measuring the similarity between the concept vector and the original inner product of the CLIP vision and text embedding. Concepts with a similarity score below a certain threshold are deemed non-interpretable.

V. To ensure the accuracy of our concept bottleneck layer in representing target concepts, we exclude concepts that do not exhibit high activation in CLIP. This activation threshold is dataset-specific, and we delete all concepts with an average top-5 activation score below the specified cutoff. Then, we can determine a cutoff threshold for each dimension of $q_i$ based on the granularity desired for the concept. A smaller cutoff value would be appropriate if we seek a more fine-grained concept set. We discard the dimensions of the concept that fall below this cutoff. This approach aids in building a sparser concept space, which proves beneficial for subsequent post-hoc analysis.

## A.2 EXPERIMENT DETAILS

We train our model using an A100 GPU (40GB). It takes less than 30 minutes to train on CIFAR-10, CIFAR-100, and STL-10 benchmarks and 8 hours to finish ImageNet training.

### A.2.1 PRETRAINED MODELS

We adopt the image encoder of CLIP and ResNet50 as our image encoder to generate concepts. Specifically, we use a pre-trained CLIP image encoder with ViT-B/16, ViT-L/14, and ResNet50 backbone. We also use ResNet50 as an image encoder pre-trained on ImageNet.

### A.2.2 TRAINING HYPERPARAMETERS

Table 4: The parameter settings of Phase 1 training: Concept project layer.

| Config | Value |
| --- | --- |
| batch size | 50000 |
| epochs | 1000 |
| optimizer | Adam |
| optimizer momentum | $\beta_1, \beta_2$=(0.9, 0.999) |
| weight decay | 0 |
| learning rate schedule | constant |
| learning rate | 1e-3 |
| early stopping | True |
| early stopping tolerance | 50 |
| feature layer | last layer (CLIP) / layer4 (ResNet50) |
| clip cutoff | 0.25 |
| seed | 42 |

Table 5: The parameter settings of Phase 2 training: VAE.

| Config | Value |
| --- | --- |
| batch size | 256 |
| maximum epochs | 450 |
| optimizer | Adam |
| optimizer momentum | $\beta_1, \beta_2$=(0.9, 0.999) |
| weight decay | 0 |
| learning rate schedule | constant |
| learning rate | 5e-5 |
| early stopping | False |
| seed | 42 |

Table 6: The parameter settings of linear probing experiments.

| Config | Value |
| --- | --- |
| batch size | 256 |
| maximum epochs | 120, 400 (SimCLR) |
| optimizer | Adam |
| optimizer momentum | $\beta_1, \beta_2$=(0.9, 0.999) |
| weight decay | 0 |
| learning rate schedule | constant |
| learning rate | 1e-3 |
| early stopping | False |
| seed | 42 |

### A.2.3 ABLATION EXPERIMENTS

In this section, we delve into evaluating our model's performance by selectively omitting different components from our pipeline. Our approach hinges on using a VAE to transform the concept

activation vector into a more compact latent space. Moreover, we intentionally incorporate class-related concepts into our concept set and utilize a combined training and testing set during VAE training. However, only the testing set is subjected to the K-means algorithm for clustering.

Table 7 offers insightful observations. One of the standout findings is that while retaining class-related concepts tends to boost performance (as seen with the CEIR(ViT-L/14) for CIFAR10, achieving NMI, ACC, and ARI values of 90.08%, 95.70%, and 90.71% respectively), omitting them does not lead to a significant decline in the metrics. This nuanced behavior is evident when comparing the performance of the CEIR(ViT-L/14) model for CIFAR10 with and without class-related concepts: the NMI drops only slightly from 90.08% to 89.35%, and similar minimal reductions are observed for ACC and ARI. This suggests that concerns about potential label leakage through class-related concepts may be unfounded, as their removal doesn't drastically affect the model's clustering ability. Further ablation studies, such as removing the test set during VAE training or solely relying on the CLIP backbone without VAE, indicate each component's integral role in achieving optimal performance. However, even in their absence, the model showcases resilience, as evidenced by the comparative scores across different datasets.

In Table 8, we investigate the impact of varying the embedding size on our VAE's efficacy. The data emphasizes the proficiency of our VAE encoder in transforming the multifaceted, high-dimensional concept embedding into a more streamlined feature space. For smaller datasets, the 128-dimensional latent embedding appears to be a more appropriate choice, capturing essential information for the classification task while concurrently filtering out extraneous noise that may compromise the latent representation. On the other hand, for larger and high-definition datasets like ImageNet, a more expansive latent embedding, notably the 256-dimensional version, proves to be more representative. This is probably because of its capacity to retain finer details inherent in such datasets.

Table 7: The results of ablation experiments.

| Datasets | CIFAR10 | | | CIFAR100-20 | | | STL10 | | |
|---|---|---|---|---|---|---|---|---|---|
| Methods (%) | NMI | ACC | ARI | NMI | ACC | ARI | NMI | ACC | ARI |
| ***Ours*** | | | | | | | | | |
| CEIR(ResNet50) | 53.27 | 69.19 | 45.31 | 44.63 | 45.09 | 28.35 | 89.71 | 95.23 | 89.87 |
| CEIR(ViT-B/16) | 81.36 | 90.46 | 80.03 | 55.63 | 57.33 | 38.80 | 96.19 | 98.48 | 96.66 |
| CEIR(ViT-L/14) | **90.08** | **95.70** | **90.71** | 65.91 | 62.53 | 48.26 | **97.87** | **99.19** | **98.21** |
| ***Remove*** *class-related concepts* | | | | | | | | | |
| CEIR(ResNet50) | 56.17 | 67.96 | 47.11 | 34.92 | 42.54 | 20.91 | 88.62 | 94.68 | 88.72 |
| CEIR(ViT-B/16) | 81.09 | 90.05 | 78.97 | 52.05 | 54.22 | 34.95 | 95.43 | 98.21 | 96.09 |
| CEIR(ViT-L/14) | 89.35 | 95.28 | 89.82 | 65.00 | 63.64 | 49.10 | 96.93 | 98.83 | 97.42 |
| ***Remove*** *test set during VAE training* | | | | | | | | | |
| CEIR(ResNet50) | 53.19 | 63.08 | 43.68 | 45.50 | 46.23 | 27.35 | 88.92 | 94.85 | 89.07 |
| CEIR(ViT-B/16) | 80.06 | 88.93 | 76.74 | 59.88 | 58.31 | 38.72 | 95.62 | 98.29 | 96.25 |
| CEIR(ViT-L/14) | 83.66 | 91.62 | 82.21 | **66.54** | **65.46** | **49.30** | 97.63 | 99.10 | 98.02 |
| ***Remove*** *VAE, CLIP backbone only* | | | | | | | | | |
| CEIR(ResNet50) | 48.62 | 54.97 | 35.57 | 36.89 | 35.75 | 18.86 | 86.09 | 89.81 | 80.81 |
| CEIR(ViT-B/16) | 75.78 | 78.25 | 68.94 | 50.84 | 48.90 | 31.72 | 93.53 | 94.93 | 90.51 |
| CEIR(ViT-L/14) | 83.67 | 83.38 | 78.21 | 49.18 | 44.17 | 30.35 | 95.12 | 95.76 | 92.09 |

## A.3 UTILIZING UNLABELED IMAGES

Incorporating additional unlabeled images into our training process allowed us to assess their impact on the model's representation learning capability. As illustrated in Table 9, the use of unlabeled data clearly augments the performance of the model in clustering tasks. Specifically, the ResNet50 backbone, which is a relatively smaller architecture, exhibited a notable improvement in clustering metrics when trained with additional unlabeled images. This suggests that for more compact models, unlabeled data can play a crucial role in enhancing their performance. On the other hand, larger architectures, such as the ViT-B/16 and ViT-L/14, also demonstrated improvements when introduced to unlabeled data. This indicates that even large models can further refine their capabilities with

Table 8: Comparison of employing different VAE latent embedding ($h$) size.

| Datasets | CIFAR10 | | | CIFAR100-20 | | | CIFAR100 | | | STL10 | | | ImageNet | | |
|---|---|---|---|---|---|---|---|---|---|---|---|---|---|---|---|
| Methods (%) | NMI | ACC | ARI | NMI | ACC | ARI | NMI | ACC | ARI | NMI | ACC | ARI | NMI | ACC | ARI |
| *Latent size 128* | | | | | | | | | | | | | | | |
| CEIR(ResNet50) | 53.27 | 69.19 | 45.31 | 44.81 | 46.93 | 26.90 | 52.45 | 36.26 | 21.55 | 89.71 | 95.24 | 89.87 | 68.14 | 38.12 | 22.41 |
| CEIR(ViT-B/16) | 81.36 | 90.46 | 80.03 | 55.63 | 57.33 | 38.80 | 67.27 | 54.90 | 38.42 | 96.19 | 98.48 | 96.66 | N/A | N/A | N/A |
| CEIR(ViT-L/14) | **90.08** | **95.70** | **90.71** | **65.91** | **62.53** | **48.26** | **78.04** | 66.77 | **54.25** | 97.87 | 99.19 | 98.21 | N/A | N/A | N/A |
| *Latent size 256* | | | | | | | | | | | | | | | |
| CEIR(ResNet50) | 54.99 | 66.80 | 45.06 | 44.63 | 45.09 | 28.35 | 48.97 | 33.13 | 19.05 | 89.14 | 95.18 | 89.71 | **78.26** | **53.09** | **38.11** |
| CEIR(ViT-B/16) | 81.31 | 90.37 | 79.85 | 57.81 | 56.37 | 40.07 | 67.14 | 54.22 | 38.64 | 97.34 | 98.94 | 97.67 | N/A | N/A | N/A |
| CEIR(ViT-L/14) | 89.87 | 95.53 | 90.37 | 64.31 | 61.98 | 46.26 | 77.09 | **67.46** | 53.90 | 97.71 | 99.11 | 98.05 | N/A | N/A | N/A |

the aid of additional data, though the magnitude of enhancement might vary. In conclusion, both compact and advanced backbones benefit from the inclusion of unlabeled images, solidifying the argument for CEIR's potential to leverage such data and possibly gain advantages from pre-training on larger datasets.

Table 9: Clustering results of CEIR trained on STL10 with/without additional 100k unlabeled images

| Backbone | Use Unlabeled Data | NMI(%) | ACC(%) | ARI(%) |
|---|---|---|---|---|
| ResNet50 | ✗ | 82.99 | 90.70 | 80.23 |
| ResNet50 | ✓ | 89.71 | 95.23 | 89.87 |
| ViT-B/16 | ✗ | 90.12 | 95.73 | 90.79 |
| ViT-B/16 | ✓ | 96.19 | 98.48 | 96.66 |
| ViT-L/14 | ✗ | 95.67 | 98.19 | 96.06 |
| ViT-L/14 | ✓ | 97.87 | 99.19 | 98.21 |

## A.4 GENERALIZATION OF CONCEPTS

Our CEIR pipeline relies on predefined concepts as a foundation for representation learning, underscoring the significance of high-quality concepts in shaping the concept bottleneck layer. Given the universality of human-derived concepts, our conceptual space demonstrates well generalizability across diverse evaluation benchmarks, as illustrated in Table 10. It's evident that a more finely-grained and systematically organized set of concepts can enhance the model's performance. However, it's worth noting that while there's a performance uptick when using more detailed concepts, the improvement is marginal for large backbone model. This capability paves the way for the development of a universal CEIR model that not only excels across different benchmarks but also performs robustly on a broad spectrum of real-world images.

Table 10: Comparison of CIFAR10 clustering performance based on different predefined concepts set. CIFAR10 indicates the concepts set generated using CIFAR10 categories. CIFAR100-20 represents the concept set generated using CIFAR100 fine classes (100) and coarser classes (20).

| Backbone | Concepts | Number of Concepts | NMI(%) | ACC(%) | ARI(%) |
|---|---|---|---|---|---|
| ResNet50 | CIFAR10 | 182 | 53.27 | 69.19 | 45.31 |
| ResNet50 | CIFAR100-20 | 1401 | **59.11** | **71.85** | **49.95** |
| ViT-L/14 | CIFAR10 | 182 | 90.08 | 95.70 | 90.71 |
| ViT-L/14 | CIFAR100-20 | 1401 | **90.21** | **95.73** | **90.74** |

## A.5 VISUALIZATION OF CONCEPTS ON BENCHMARKS

This section displays the concepts mined from our CEIR pipeline for each image, depicted in Figure 7, 8, 9, 10, 11, 12. The input images are randomly sampled from STL10, CIFAR10, and CIFAR100

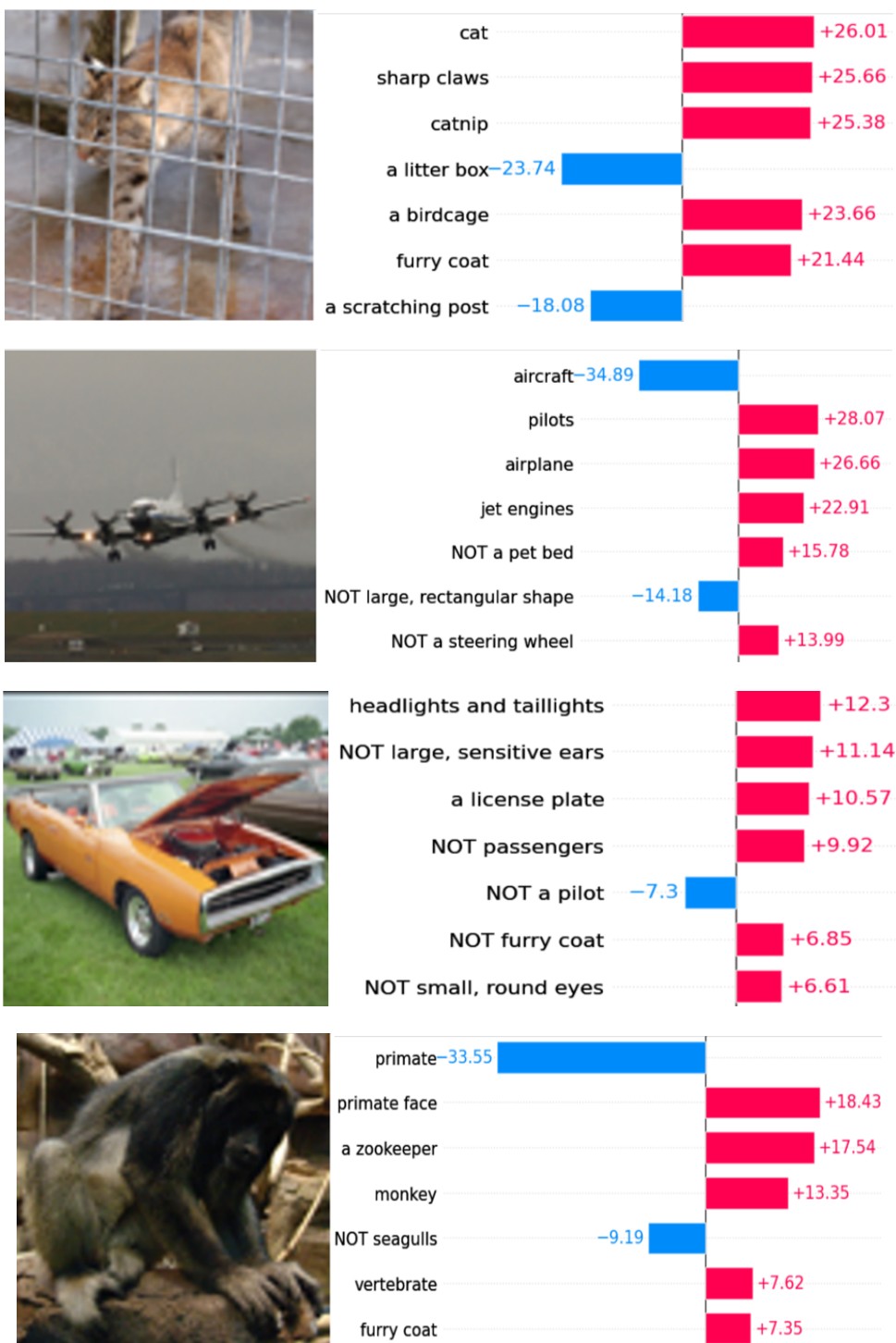

Figure 7: dataset: STL10, from top to bottom, the class for each image is 'cat'; 'airplane'; 'car'; 'monkey';

benchmarks, and for each dataset, we select four images to show its concept from our model. We can see that the concept bottleneck layer can provide a good description of each image on an unsupervised manner. The backbone model is CLIP-ViT-L/14, and the loaded models are trained on each benchmark respectively.

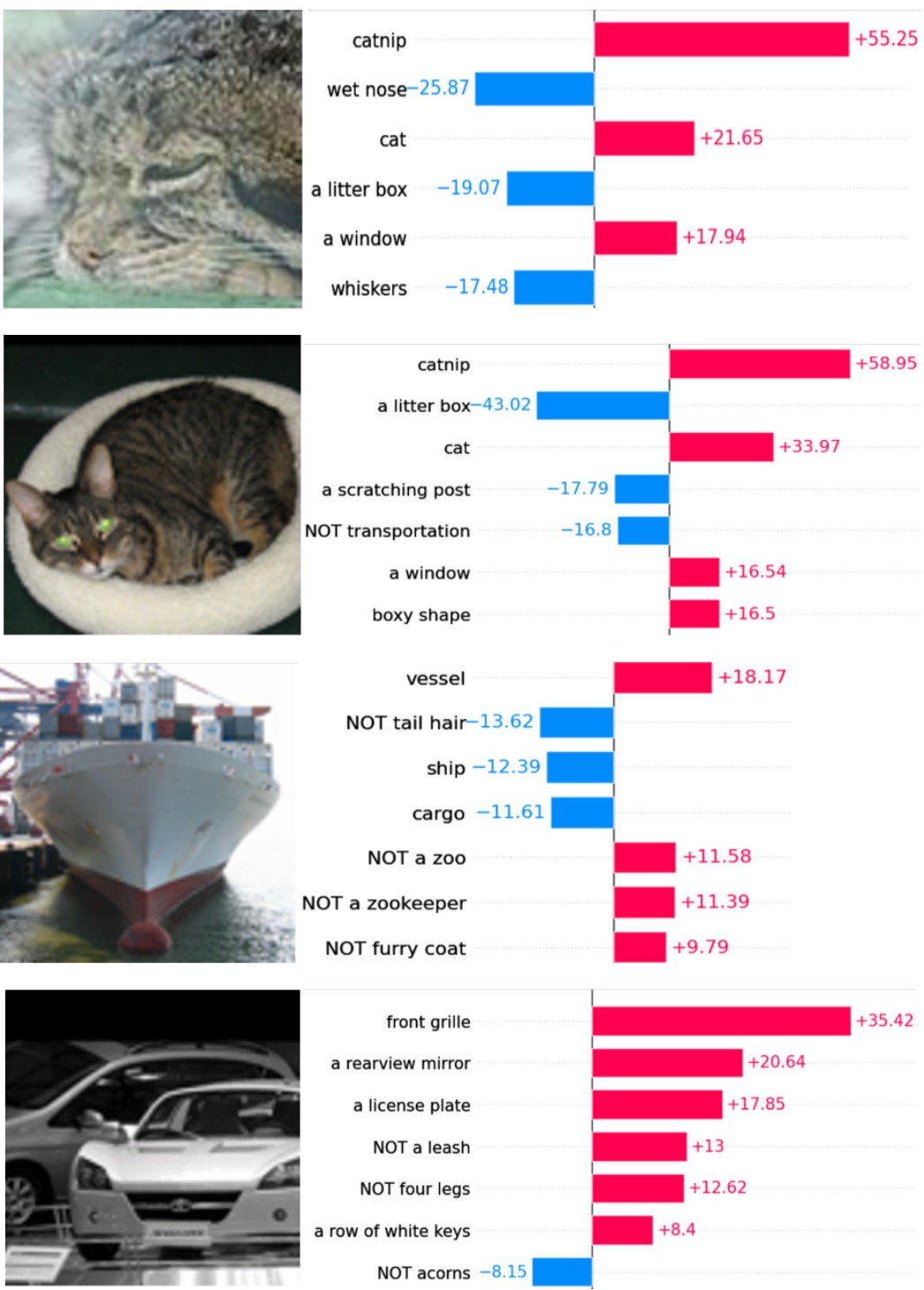

Figure 8: dataset: STL10, from top to bottom, the class for each image is 'cat'; 'cat'; 'ship'; 'car';

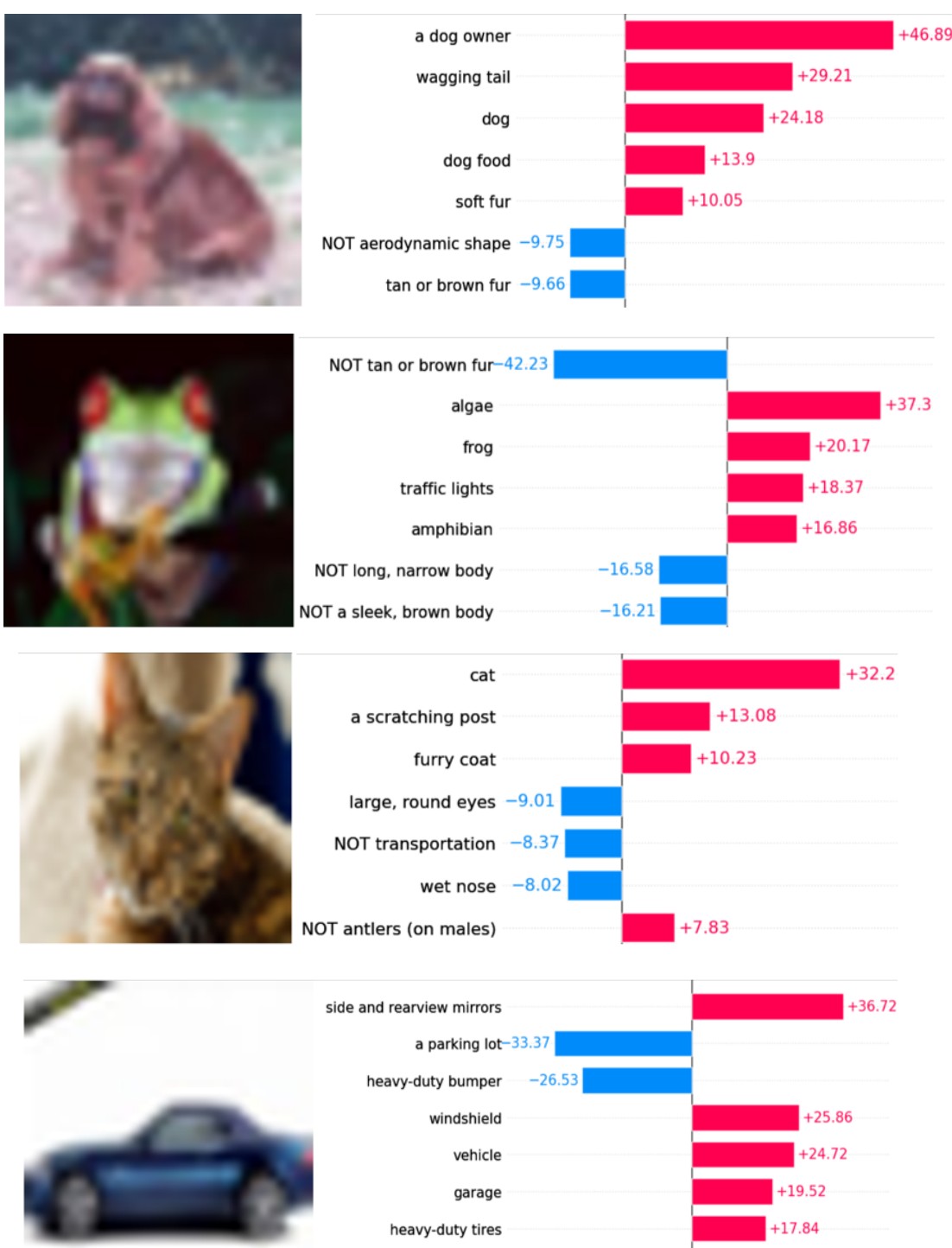

Figure 9: dataset: STL10, from top to bottom, the class for each image is 'dog'; 'frog'; 'cat'; 'automobile';

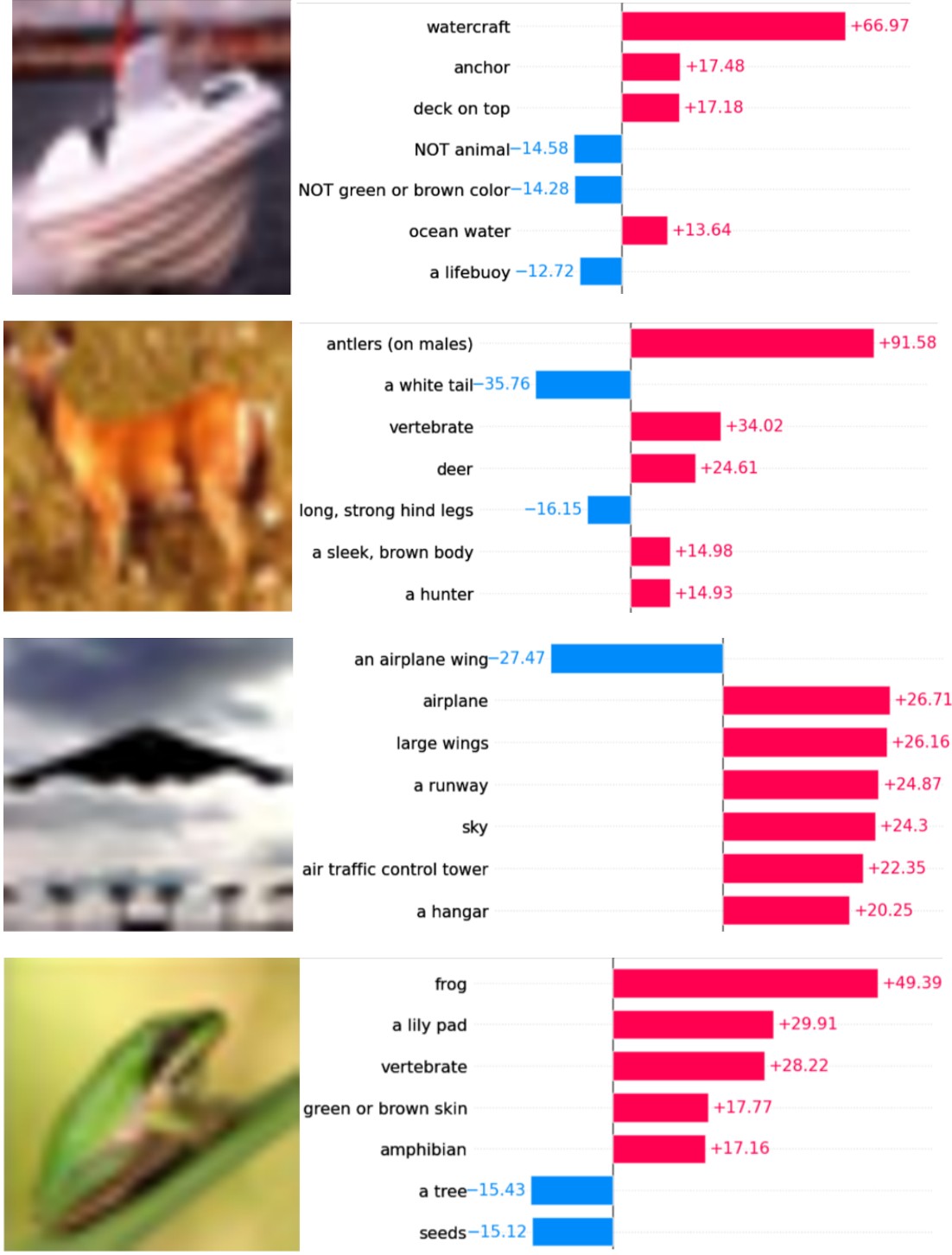

Figure 10: dataset: cifar10, from top to bottom, the class for each image is 'ship'; 'deer'; 'airplane'; 'frog';

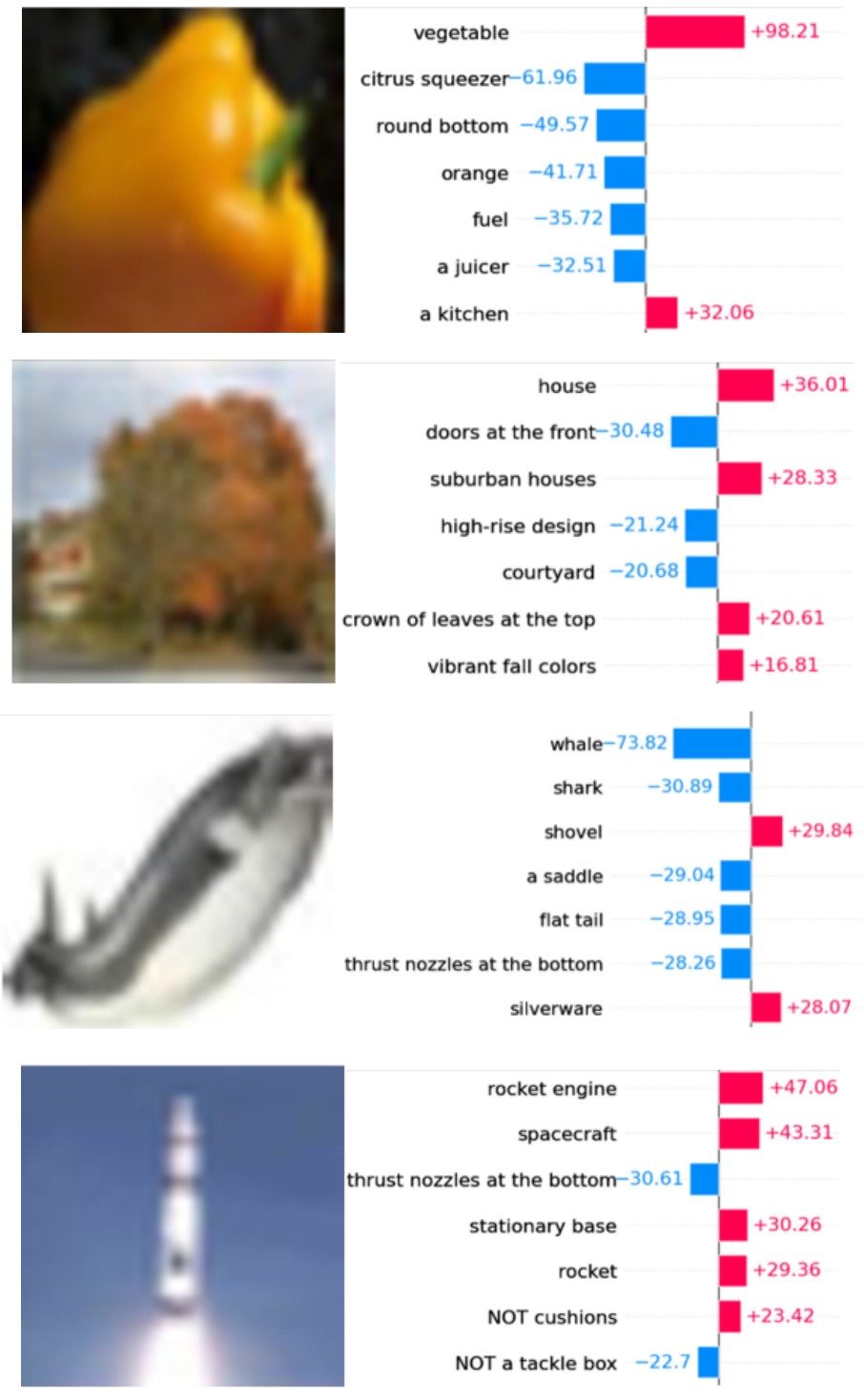

Figure 11: dataset: cifar100, from top to bottom, the class for each image is 'sweet pepper'; 'maple tree', 'whale'; 'rocket';

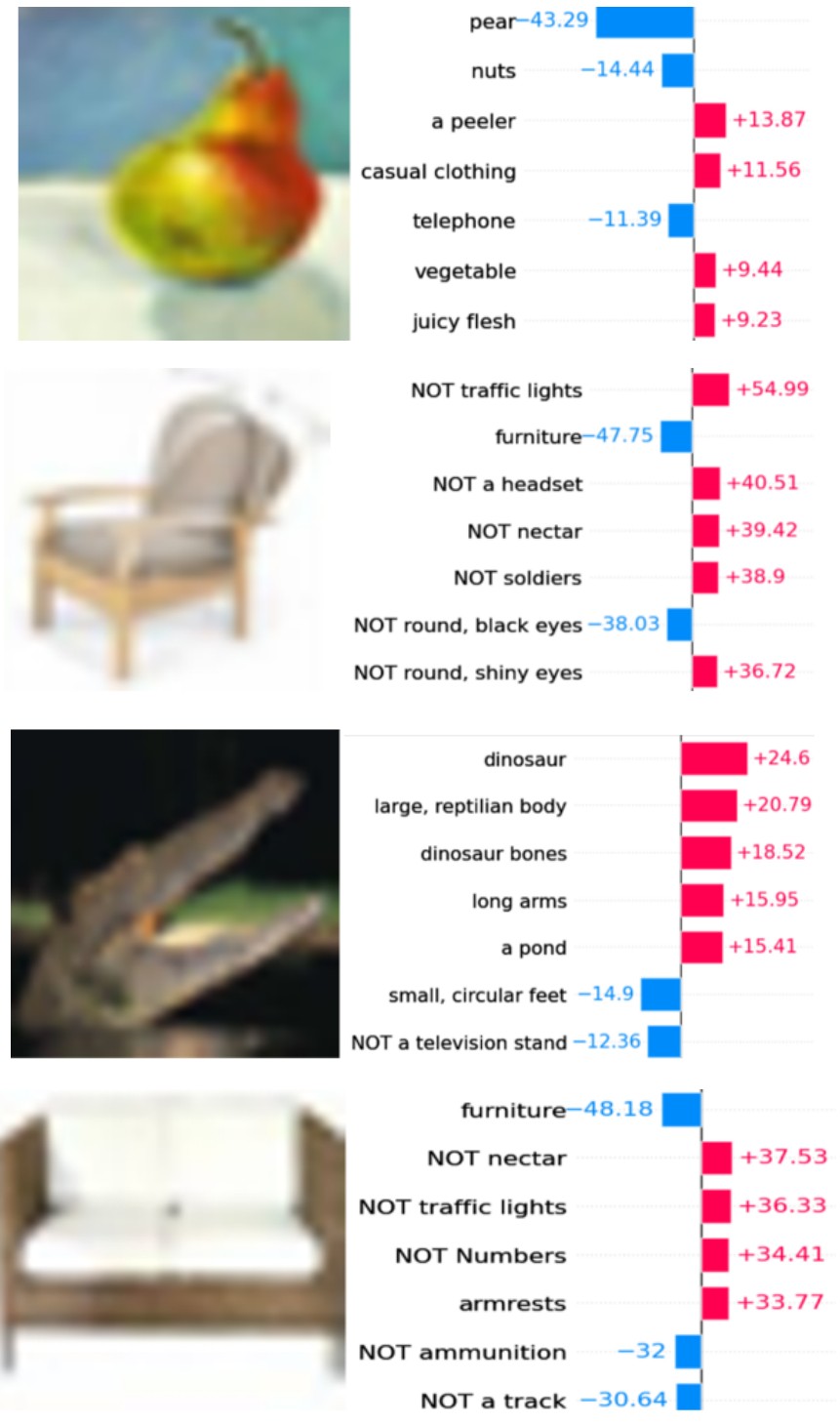

Figure 12: dataset: cifar10, from top to bottom, the class for each image is 'pear'; 'chair'; 'crocodile';'coach'

## A.6 VISUALIZATION OF CONCEPTS ON REAL-WORLD IMAGES

In this section, we capture real-world images using a smartphone to demonstrate the capacity of CEIR to discern and extract concepts from arbitrary images. The backbone image encoder employed is ResNet50, trained on ImageNet. As shown in Figures 13 and 14, our model can recognize and derive concepts from unseen real-world images.

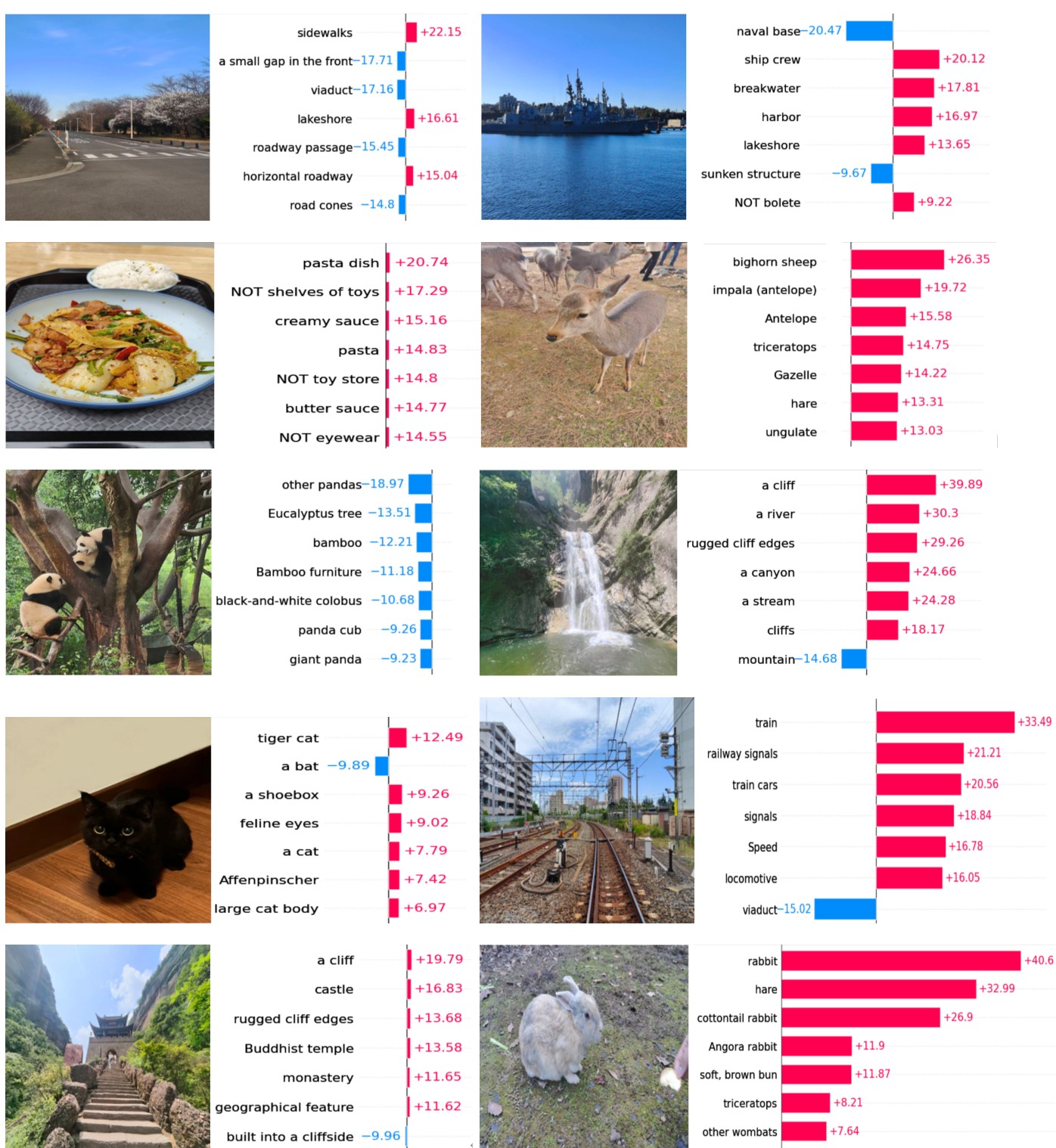

Figure 13: Photos taken from the real world and their concepts revealed by CEIR pipeline.

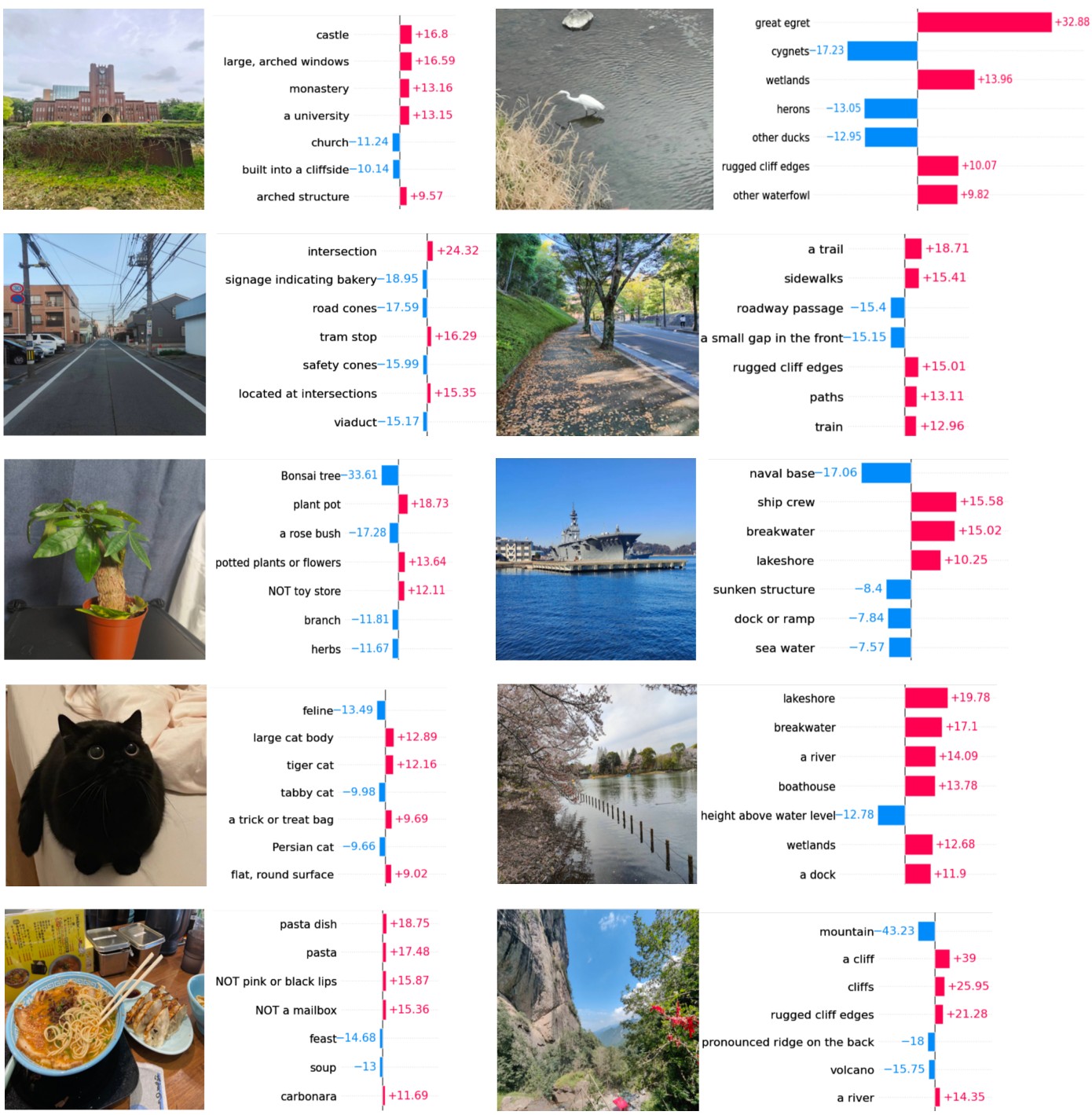

Figure 14: Photos taken from the real world and their concepts revealed by CEIR pipeline.

