# OpenReview forum: "CEIR: Concept-based Explainable Image Representation Learning"
_ICLR.cc/2024/Conference — Submitted to ICLR 2024_

### Official Review · Reviewer_mkN4 · 2023-10-23

**Soundness:** 1 poor
**Presentation:** 2 fair
**Contribution:** 1 poor
**Rating:** 3
**Confidence:** 4

**Summary:**

The authors propose to build an image representation as follows:
- build a set of (textual) concepts with GPT4
- learn a projection that aligns the output of a backbone with the similarity matrix between the CLIP representation of an image and the CLIP encodings of the concepts.
- train a VAE to reduce this representation
- use the reduce representation for some clustering tasks
- interpret the compressed concepts with Crabbe & van der Schaar (2022)

**Strengths:**

Representation learning is a topic that remains relevant within the computer vision community.

**Weaknesses:**

It seems to me that the main weakness of this paper lies in a form of misunderstanding. The authors pretend that their work is an unsupervised method (representation learning), and the whole motivation of the paper comes from this, , all the way down to the clustering task used to validate the method.

The reality is in fact quite different, in that the core of the method is based on CLIP, which has been learned from a gigantic database of annotated (captioned) images. CLIP's zero-shot classification performance is very good, as numerous papers have shown. The CLIP representation is likely to contain precise definitions of the database labels used to evaluate the proposed method. Not surprisingly, a representation derived from CLIP is good at doing clustering. The problem is that evaluating a method on a clustering task when the method uses labels is not fair at all.

**Questions:**

I have no particular question, apart from the one mentioned above, about the control that the use of CLIP introduces into the proposed representation.

---

### Official Review · Reviewer_8cNt · 2023-10-25

**Soundness:** 1 poor
**Presentation:** 1 poor
**Contribution:** 2 fair
**Rating:** 3
**Confidence:** 4

**Summary:**

The paper presents a method for concept-based image representation, where the set of concepts is obtained by using GPT-4. As far as I understand, the concept extractor is trained by distilling CLIP through aligning the linear projection of an input image to the concept space with the similarities between the input image and textual representation of the concept. The dimensionality of the concept vectors (an input image after the linear projection) is then reduced using VAE. The experimental results provide some comparison.

**Strengths:**

(1) The idea may be interesting but I don’t understand the method and I’m not very sure about this.

**Weaknesses:**

(1) I cannot understand what the method is doing. For example, I don’t see what the superscript “3” in Eq. (1) means. Does it mean cubic? Why it’s necessary? $\mathcal{Q}$, $\mathcal{H}$, $\mathcal{A}(\mathbb{R}^\mathcal{Q})$, and $\mathbb{R}$ are not defined (at least the paper is not self-contained). All these missing details make it almost impossible to see what’s going on in the method.

(2) My major question is the reason why the backbone and the projection layer are necessary. If $P_{i,:}$ is the target when training the projection layer and $q_i$ is expected to be closer to $P_{i,:}$, one can use $P_{i,:}$ as a concept vector.

(3) If I understand correctly, VAE compresses the concept vectors to a latent vector. I don’t see why this is necessary. What is the typical number of concepts for each dataset (are they like 182 or 1401 as in Table 10)? As all the information provided in the latent vector is in the concept vector, I’m not sure applying dimensionality reduction really benefits some aspects of the method. An ablation study may help understand.

(4) I cannot get what is evaluated in the experiment section. I think the paper should evaluate how well the method can find the designated concepts in the concept vectors. The method has two concept representations (concept vectors and concept importance), so their consistency should be evaluated. I understand that this is not straightforward, as there are no annotations on the concepts, but I think evaluation over a small subset is helpful. Also, the reference set for each class should be evaluated in some way (I’m sorry, I didn’t come up with a good way except for human evaluation) to show the validity of the approach. I also think the representation should be evaluated for downstream tasks, like classification.

**Questions:**

I would like to see some discussion on (2) and (3).

---

### Official Review · Reviewer_Xytc · 2023-10-31

**Soundness:** 3 good
**Presentation:** 1 poor
**Contribution:** 2 fair
**Rating:** 3
**Confidence:** 4

**Summary:**

This paper proposes a method to generate concept-based explanations for image classifiers. The method uses GPT and CLIP to identify a set of nameable visual concepts, then projects the image embeddings from the classifier into the CLIP concept space to get a concept-based representation of the images. This representation is further simplified by using an autoencoder to project the rather large set of concepts into a smaller set that represents only the most important concepts. The method is evaluated as an unsupervised learning method and produces clustering results comparable to or better than state-of-the-art on ImageNet, CIFAR, and STL-10 datasets.

**Strengths:**

The method produces impressive clustering results on ImageNet, CIFAR, and STL-10 datasets (comparable to or above state of the art).

Using CLIP to find nameable concepts for XAI is a good idea, and the paper demonstrates how this makes it easier to access and interact with the concepts (e.g., find more images from another class that contain the same concept as a given class).

**Weaknesses:**

The writing is frequently unclear, which makes many parts of the paper hard to understand.

The evaluation is limited to everyday object/scene datasets (ImageNet, CIFAR, STL-10) which are the datasets where this approach should work best due to the high overlap with CLIP’s training set. It would be nice to see evaluation on a broader range of datasets.

There's no user experiment, so it's unclear if these explanations would be useful for humans or how they compare to other XAI approaches.

The explanations don't seem to be entirely correct, given the examples in Figure 3. It seems like the model just lists concepts that are related to the predicted class, regardless of whether they are present in the image (e.g., “lion-like mane” for a female lion, or “sun lotion” for an image mistaken for a swimsuit photo, or “rotational movement” for an image containing a ball).

**Questions:**

How would this approach work with domain-specific classifiers, like ones designed for medical images (skin cancer detection, retinal image analysis, etc.), or even a more specific object classifier like Caltech-UCSD-Birds (CUB)?

Do the concepts actually reflect things that the model thinks are in the image, or are they just things associated with the predicted class (for example, why does the model detect "sun lotion" in the top image of Figure 3)?

Why were the transformer-based models not used with ImageNet in the evaluation?

Is it appropriate to include the test images when training the VAE? This seems to make the evaluation less reliable, since the model can learn a good representation for the test images, instead of having to encode the test images into a representation that’s based on the training images.

What does the sign indicate on the right side of Figure 3? It’s not explained and I assume it’s irrelevant (if so, this figure would be clearer if it showed absolute values).

In 4.4, how accurate was this approach (how well did it link the 24 images to the ~40 identified concepts)? This is hard to evaluate from Figure 4.

As a side note -- there appears to be an error in the citation formatting which causes citations to run into the text (e.g., "pretrained models such as CLIP Radford et al. (2021)").

---

### Official Review · Reviewer_XinF · 2023-11-02

**Soundness:** 2 fair
**Presentation:** 2 fair
**Contribution:** 2 fair
**Rating:** 3
**Confidence:** 4

**Summary:**

This paper proposes CEIR, a new unsupervised approach to learn concept-based image representation and thus representations that are more interpretable by humans. The concepts are first generated with a prompting strategy using GPT-4. Then concept vectors are built leveraging CLIP and the well-known concept bottleneck model. CLIP serves here as a kind of supervision. Then, a VAE is used to learn a reduced-size representation. The paper proposes a large experimental study to evaluate the proposed representation using clustering and linear probing with also a comparison to the state of the art. The proposed representation provides new state-of-the-art results for clustering. Other experiments also concern the explainability part of the learned representation and the unsupervised part. A large set of appendices presents details on these experiments.

**Strengths:**

+ The paper addresses an important issue in representation learning: the ability to learn human-understandable representation without the need for a large annotated dataset.
+ The proposed idea is simple, leveraging different existing models: clip-based models, text generative models, VAE. It is a nice way to combine existing ideas in the field of representation learning and XAI.
+ The proposed approach enables state-of-the-art results on clustering tasks on different visual classification benchmarks.
+ A large experimental study is provided including an ablation study on the size of the learned representation.

**Weaknesses:**

I have several concerns :
+ My first concern is related to the positioning of the paper compared to concept-based explainable image representation. In particular, in the XAI field, some criteria and properties have been proposed to define the concept of good explainable representation (see for instance the work of Ghorbani [here](https://arxiv.org/pdf/1902.03129.pdf)) such as meaningfulness, coherency, and importance. How the proposed approach tackles these aspects is not clear and not evaluated at all. In particular, the proposed approach is not compared to the existing state-of-the-art on concept-based explainable image representation, supervised or unsupervised.
+ In the same vein, since the claim of the paper is to reach to human understandable image representation, an experimental study that supports this claim should be provided. In particular, some metrics (correctness, stability, plausibility) have been proposed in the XAI field to evaluate explanations without human-level studies. See for instance [this paper](https://arxiv.org/abs/2303.15632) for faithfulness and understandability criteria.
+ Some components of the proposed approach lack a clear justification and motivation for instance the GPT part compared to the use of existing explicit knowledge (e.g. Wordnet ontology or existing knowledge graphs).
+ Some technical details are missing to evaluate the results correctly. For instance, in Table 2, what is the size of the corresponding representations ?
+ Another concern is related to written level of the paper. The paper needs a complete proofreading since it contains a lot of mistakes (bad punctuation, strange ways to indicate the references).
+ Sometimes some references given are not good ones. For instance, in the introduction, Yang et al and Crabbé are not seminal works for concept-based representations.

**Questions:**

+ In relation to my concerns, what are the properties that should handle the targeted concept space?  How to assess them?
+ Why a process to generate concepts rather than selecting predefined human-defined concepts? Appendix 4 gives some first insights but since large language generative models are not transparent and are themselves black-box models, why add this step in the whole process? For instance, leveraging existing common-sense knowledge graphs could be a better alternative.
+ The idea of building a latent representation of the concept vector is original but it leads to a loss of the interpretability of the obtained representation. There is a large body of literature that used VAE in representation learning for their ability to learn a kind of disentangled representation.  What about this aspect in the proposed scheme?
+ Explainability results given in the paper are prone to a large discussion. For instance, how to interpret human negative concepts ? How do the concepts relate to the image? Some approaches propose to visualize concepts through prototype visualization for instance or by activation maps in the image.
+ Globally, some ability tools mentioned for the proposed approach should be compared to alternatives. For instance, for section 4.4 how to compare in terms of quality but also time to build the obtained datasets compared to other approaches to build them?

---

### Meta-Review · Area_Chair_yfLJ · 2023-12-10

**Metareview:**

The submission studies interpretable image representation learning. All four reviewers raised significant concerns about the presentation, evaluation, and presentation of the submission. There is no rebuttal. The AC agrees with the reviewers and recommends rejection.

**Justification For Why Not Higher Score:**

There are significant concerns about the presentation, evaluation, and presentation of the submission.

**Justification For Why Not Lower Score:**

N/A

---

### Decision · Program_Chairs · 2024-01-16

Reject